# Effects of Genetic Variation of the Sorting Nexin 29 (*SNX29*) Gene on Growth Traits of Xiangdong Black Goat

**DOI:** 10.3390/ani12243461

**Published:** 2022-12-08

**Authors:** Yuhan Chen, Long Yang, Xiaoding Lin, Peiya Peng, Weijun Shen, Sipei Tang, Xianyong Lan, Fachun Wan, Yulong Yin, Mei Liu

**Affiliations:** 1Animal Nutritional Genome and Germplasm Innovation Research Center, College of Animal Science and Technology, Hunan Agricultural University, Changsha 410128, China; 2Liuyang Agricultural and Rural Bureau of Hunan, Liuyang 410300, China; 3Shaanxi Key Laboratory of Molecular Biology for Agriculture, College of Animal Science and Technology, Northwest A & F University, Xianyang 712100, China

**Keywords:** *SNX29*, indel, copy number variation, goats, growth traits

## Abstract

**Simple Summary:**

With the development of goat industry, the improvement of goat production performance has become the primary task of goat industry. Studies have shown that *SNX29*, known for binding to microtubule motor protein activity and phosphatidylinositol, could be a candidate gene for carcass and dorsal fat traits. Many studies have been provided that CNV and indel can affect their phenotypic traits in livestock. However, there are no reports on investigating *SNX29* genetic variation in Hunan local goats. In this study, two indels (17 bp-indel and 21 bp-indel) and CNV of *SNX29* were identified in 516 individual Xiangdong black goats (a representative Chinese native goat breed in Hunan), while *SNX29* CNV was significantly associated with chest circumference and abdominal circumference (*p* < 0.01). These findings suggested that the *SNX29* gene could be used for marker-assisted selection in meat goats. For the first time, this report reveals a novel application prospect for the *SNX29* CNV to improve the Chinese local goats.

**Abstract:**

Previous studies have found that the copy number variation (CNV) and insertion/deletion (indels) located in the sorting nexin 29 *(SNX29*) gene, which is an important candidate gene related to meat production and quality, are associated with growth traits of African goats and Shaanbei white cashmere goats. However, the genetic effects of *SNX29* genetic variation on growth traits of Xiangdong black (XDB) goat (a representative meat goat breed in China) are still unclear. The purpose of this study was to detect the mRNA expression level of *SNX29* and to explore the genetic effects of CNV and indel within *SNX29* on growth traits and gene expression in XDB goat. The *SNX29* mRNA expression profile showed that the *SNX29* was highly expressed in adipose tissues, indicating that the *SNX29* gene could play a key role in subcutaneous adipose deposition of XDB goat. 17 bp indel (g.10559298-10559314), 21 bp indel (g.10918982-10919002) and CNV were detected in 516 individuals of XDB goat by PCR or qPCR. The association analysis of *SNX29* CNV with growth traits in XDB goats showed that *SNX29* CNV was significantly correlated with chest circumference and abdominal circumference (*p* < 0.01), and the normal type of *SNX29* CNV goat individuals were more advantageous. For the mRNA expression of *SNX29* gene, individuals with *SNX29* copy number normal type had a higher trend than that of *SNX29* gene with copy number gain type in longissimus dorsi muscle (*p* = 0.07), whereas individuals with *SNX29* copy number gain type had a higher trend in abdominal adipose (*p* = 0.09). Overall, these results suggested that the *SNX29* gene could play an important role in growth and development of XDB goats and could be used for marker-assisted selection (MAS) in XDB goats.

## 1. Introduction

The Xiangdong black (XDB) goat is a representative Chinese local meat goat breed with pure black coat. It has the characteristics of minimal smell of mutton, delicious taste and high nutritional value. Currently, XDB goats are mainly distributed in the eastern mountainous areas of Hunan Province. Compared with foreign meat goat breeds, XDB goats still had many shortcomings, such as slow growth and low meat yield. Many studies have shown that individual phenotypic variations are result from the genomic DNA variation [1,2]. Genomic variation could be, according to their size, classified as single nucleotide polymorphism (SNP, i.e., single base change), insertion and deletion (indel, i.e., small structural variation with size ranging from 2 nt to 50 nt), structural variation (SV, i.e., large structural variation with size >50 nt). Essentially, the later two types are genomic structural variation. Structural variation also contains chromosomal inversion. Copy number variation (CNV) is a type of structural variation, reflecting duplication (copy number gain) and deletion (copy number loss) [3,4,5,6]. Evidences have been provided that SV can affect their phenotypic traits in livestock such as pigs and cattle [6,7,8,9]. Also, SV could play important roles in animal diseases, species evolution, and genetic breeding [10,11,12]. Up to now, the research on functional gene variations in XDB goats was still lacking. Therefore, it is necessary to explore the candidate functional gene variations that are related to meat production and quality of XDB goats.

The sorting nexin 29 (*SNX29*), a member of the sorting nexin (SNX) family, is a protein coding gene which can promote myoblast differentiation by combing microtubule motor protein activity with phosphatidylinositol [13,14]. Studies have shown that overexpression of circSNX29 directly binds to miR-744 to increase the expression of Wnt5a and CaMKIIδ and initiate the Wnt5a/Ca^2+^ signaling pathway, thereby promoting the proliferation and differentiation of bovine myogenic cells [13]. In Soay sheep, three immune traits were studied by genome-wide association studies (GWAS). It was found that there was a highly related SNPs on chromosome 24 of adult sheep, which corresponds to the *SNX29* gene [15]. High-throughput sequencing of circRNAs from the skeletal muscle of Qinchuan cattle at different growth and development periods revealed that circRNA243 derived from the *SNX29* gene was significantly down-regulated during skeletal muscle production [16]. Dong et al. [17] detected the meat quality traits of pigs by 60 k SNP chip and found that SNP in *SNX29* was related to the content of intramuscular fat. In our previous study, CNV detection was carried out on African meat goats, which showed that CNV27, which was overlapped with the *SNX29* gene, was significantly correlated with growth traits such as the chest width of goats [18]. Recently, the work in our group have shown that *SNX29* indels were associated with growth traits in Shaanbei white goats [19]. These results suggest that *SNX29* gene variations could be related to meat quality and productive traits of livestock.

However, there are no reports on the investigation of *SNX29* gene variations distribution and their genetic effects on phenotypic traits in XDB goats up to now. Thus, this study aimed to explore the possible effects of SV of goat *SNX29* gene on growth traits of XDB goats. These results will help us to further investigate the relationship between growth traits in livestock. Meanwhile, we expect it could provide a theoretical basis for the genetic improvement of goat.

## 2. Material and Methods 

The experiments carried out in this study was approved by the International Animal Protection and Utilization Committee of Hunan Agricultural University (protocol number: 20200901), and complied with local laws and policies on animal welfare.

### 2.1. Bioinformatics Analysis

The amino acid sequences of the SNX29 protein of *Homo sapiens* (XP_047290835.1), *Sus scrofa* (XP_020943714.1), *Bos taurus* (XP_024840650), *Bos indicus* (XP_019843531.1), *Bos mutus* (XP_005889329), *Capra hircus* (XP_017896122.1), *Ovis aries* (XP_042096118.1), *Bubalus bubalis* (XP_044791693.2), *Rattus rattus* (XP_032768746.1), *Mus musculus* (NP_001390164.1), *Gallus gallus* (XP_046783829.1) were downloaded from NCBI database (https://www.ncbi.nlm.nih.gov/protein (accessed on 5 May 2022)). Multiple sequence comparisons were performed using the MEGA X 10.2.6 MUSCLE program and Neighbor-Joining (NJ) method to construct phylogenetic trees [20]. To analyze the structural featural and functions of SNX29 proteins, the motifs of SNX29 proteins among different species was investigated by the MEME suite (https://meme-suite.org/ (accessed on 5 May 2022)) [21]. The conservative domains and their functions were analyzed by CDD from NCBI (https://www.ncbi.nlm.nih.gov/cdd/ (accessed on 5 May 2022)) [22].

### 2.2. Samples and Phenotypic Data Collection

Under the same breeding conditions, 516 adult non-pregnant female goats, which were about 2 years old (730 ± 60 days) were randomly selected from Xiangdong black goat conservation farm. Blood sample was collected from all goats. Meanwhile, those goat growth characteristics were measured, including body height (BH), body length (BL), body oblique length (OL), chest circumference (CC), abdominal circumference (AC), cross height (CH), and cannon circumference (CAC). Different tissues, including heart, spleen, lung, kidney, small intestine, large intestine, stomach, subcutaneous adipose, abdominal adipose, longissimus dorsi muscle, leg muscle, of three adult female goats were collected for gene expression analysis experiments. All samples were frozen immediately in liquid nitrogen after collection and then stored at −80 °C.

### 2.3. Total RNA Isolation, DNA Extraction and Primer Design

The genomic DNA was extracted from blood by phenol chloroform method [23], the concentration of the samples was assayed by a Nanodrop one spectrophotometer (Thermo Fisher Scientific, Waltham, MA, USA), the DNA was diluted to the standard concentration of 20 ng/µL and stored at −20 °C. Total RNA was extracted using Trizol (Takara, Japan) according to the manufacturer’s instructions. Then, RNA was synthesized to first-strand cDNA using a PrimeScript^TM^ RT reagent Kit with gDNA Eraser (Takara, Kusatsu, Japan) according to the instruction manufactory.

The sequence of *SNX29* gene in goat (NC_030832.1) in the NCBI database was used as the reference sequence, and the CNV was located in Chr25: 10572059-10683032 [18]. The primers were designed to detect CNV based on the *SNX29* DNA sequence. The melanocyte-stimulating hormone receptor (*MC1R*) gene was used as internal reference gene [24]. The primers for detecting indels were derived from previously published article [19]. Then, according to the sequence of *SNX29* (XM_018040633.1), the primers for quantitative real-time PCR (qPCR) were designed, and the *GAPDH* (XM_005680968) gene was selected as the internal reference gene. Primers’ information was shown in Appendix A. All primers were designed by Primer Premier 6.0 software (Premier, Vancouver, BC, Canada). The specificity and dissolution curve of the primers needed in the experiment were evaluated, and the effective primers were used in the subsequent tests.

### 2.4. Copy Number Analysis and mRNA Expression of SNX29 Gene

In this study, the copy numbers and mRNA expression level of *SNX29* gene in goats were examined. The genomic DNA from blood and cDNA from tissues were used separately for qPCR. The qPCR experiment at DNA and mRNA levels were performed with three repeated reactions on SYBR^®^Green. The 10 µL reaction system contained 5 µL SYBR^®^Premix Ex Taq II, 1.5 µL genomic DNA or cDNA, 1 µL each of forward and reverse primers (the primers sequence were reported in Appendix A) and 1.5 µL ddH_2_O. The experimental procedures were as follows: 95 °C for 45 s, 45 cycles for 15 s, 5 s at 95 °C and 40 s at 60 °C.

### 2.5. Polymerase Chain Reaction Amplification and Indel Genotyping of the SNX29 Gene

To identify variants, polymerase chain reaction amplification (PCR) was performed using genomic DNA from blood. PCR experiments were performed on all genomic DNA for indel genotyping of the *SNX29* gene. The reaction system included 3 µL genomic DNA, 0.5 µL each of forward and reverse primers (the primers sequence were reported in Appendix A), 6 µL ddH_2_O and 10 µL 2 ×Taq PCR MasterMixⅡ. The reaction conditions were as follows: 94 °C for 3 min; followed by 35 cycles of 30 s at 94 °C, 30 s at 55 °C, and 1 min at 72 °C; the product was kept intact at 72 °C for 5 min and cooled at 4 °C. Finally, to identify 17 bp indel and 21 bp indel genotypes, the PCR products were genotyped by 3% agarose based on the band size.

### 2.6. Statistical Analyses

Genetic homozygosity (HO), genetic heterozygosity (HE), polymorphic information content (PIC) and effective allele number (Ne) were calculated the gene genetic parameters according to the genotyping results. The Hardy–Weinberg equilibrium (HWE) was detected by SHEsis program (http://analysis.bio-x.cn (accessed on 15 March 2022)). In addition, the R package (LD heat-map) was used to calculate the pairwise linkage disequilibrium (LD) between the two indel sites.

The copy number of the *SNX29* gene in this study was calculated by 2 × 2^−ΔΔCt^ (ΔCt = Ct_SNX29-CNV_–Ct_MC1R_) [25]. According to formula 2 × 2^−ΔΔCt^, the copy number variation of *SNX29* gene was divided into three types, loss (<2), normal (=2) and gain (>2). The relative differential expression levels of *SNX29* gene in different tissues were detected by 2^−ΔΔCt^ method (ΔCt = Ct_SNX29_–Ct_GAPDH_) [25]. All tests were repeated three times and then the mean value ± standard error (SE) was plotted [6]. The expression of *SNX29* gene was detected at mRNA level by Excel 2010 and prism 6.0 software. The correlation between CNV/indel genotypes within *SNX29* and growth traits of XDB goats were analyzed using SPSS V26.0 (SPSS, Inc, Chicago, IL, USA) software by one-way analysis of variance (ANOVA) method. The following linear model was established: Y_i_ = µ + G_i_ + e, where Y_i_ was the trait measured by each goat, µ was the overall mean for each growth trait, G_i_ was the effect of fixed factor indel genotypes/copy numbers and e was the random error. The correlation analysis of CNV and mRNA expression levels of *SNX29* were performed by *t*-test.

## 3. Results

### 3.1. Biological Evolution and Conservation Analysis of SNX29

To initially explore the function of *SNX29* gene, the amino acid sequences of SNX29 proteins from 11 species were analyzed for comparison, including human (*Homo sapiens*), pig (*Sus scrofa*), cattle *(Bos taurus*), zebu cattle *(Bos indicus*), yak (*Bos mutus*), goat (*Capra hircus*), sheep (*Ovis aries*), buffalo (*Bubalus bubalis*), rat (*Rattus rattus*), mouse (*Mus musculus*), and chicken (*Gallus gallus*). The protein structure was found to be highly conserved in cattle, zebu cattle, yak, buffalo, goat, sheep and pig, and slightly different in rat, mouse, human and chicken (Appendix A). Seven significant motifs were found in the SNX29 protein of eleven species (Appendix A). Meanwhile, the phylogenetic tree analysis showed that cattle, buffalo, yak, goat, sheep and pig were more closely related, while human, rat, mouse and chicken were more distantly related to goat (Appendix A). NCBI CDD was used to analyze the SNX29 protein structure, four specific conserved domains, RUN (cl45896), ZIP_TSC22D-like super family (cl40461), DUF460 super family (cl43687) and PX_RUN (cd07277), were found in eleven species (Appendix A). These results indicated that the SNX29 protein was highly conserved and could play a consistent role in eleven species. Besides, the *SNX29* CNV motif and conserved regions were shown in Appendix A, which the sequence of *SNX29* CNV was significantly associated with ZIP_TSC22D-like super family and DUF460 super family.

### 3.2. Distribution of Copy Numbers Variation and Indels of SNX29 in XBD Goats

In this study, we detected the copy number of *SNX29* gene in XDB goat population by qPCR. The distribution of *SNX29* copy number variation in XBD goat population was shown in Figure 1A, where the *SNX29* copy number ratio mainly showed 1 to 6 copies in XDB goat population. As shown in Figure 1B, the proportion of normal type copy number in XDB goats was the highest (59%), and the copy numbers of loss type and gain type were less distributed in this population, 17% and 24% respectively.

Similar to the DNA pools method in our previous study [19], two indels loci, 17 bp indel (g.10559298-10559314) and 21 bp indel (g.10918982-10919002), were found to be polymorphic in XDB goats (Appendix A). Genotyping of *SNX29* gene indels was performed by PCR. The 17 bp indel and 21 bp indel generated three genotypes: Homozygote insertion (II), homozygote deletion (DD) and heterozygote (ID) (Appendix A).

### 3.3. Population Genetics Analysis for Two SNX29 Indels in XBD Goats

According to the genotyping results in goats, the genotypic distribution, allelic frequencies and genetic parameter of the two indels loci were showed in Appendix A. For 17 bp indel, the dominant allele was I, with a frequency of 0.847. For 21 bp indel, the dominant allele was I, with a frequency of 0.565. Besides, 17 bp indel and 21 bp indel were in Hardy-Weinberg equilibrium (HWE) (*p* > 0.05). The genetic diversity was divided according to PIC values, the 17 bp-indel displayed low genetic diversity (PIC < 0.25), and 21 bp-indel displayed medium genetic diversity (0.25 < PIC < 0.50). In addition, no strong linkage was found between the two indels loci (r^2^ < 0.33) (Appendix A).

### 3.4. Association Analysis for the Genetic Variations of SNX29 and XBD Goat Growth Traits 

In XDB goats, we analyzed the association between the *SNX29* gene CNV and growth traits of XDB goats by the general linear model. The analysis showed that *SNX29* CNV was significantly correlated with CC and AC traits (*p* < 0.01), and the CC and AC traits of normal *SNX29* CNV goats were better than those goats with loss and gain *SNX29* CNV types (Table 1). However, the association analysis found that the two indel loci (17 bp indel and 21 bp indel) were not associated significantly with growths traits of goats (*p* > 0.05) (Appendix A).

### 3.5. Analysis for the SNX29 Gene Expression and Its Associations with the SNX29 CNV in XBD Goats 

The *SNX29* mRNA expression was detected in eleven tissues of XDB goats. As shown in Figure 2A, the *SNX29* mRNA was widely expressed in adult goat tissues. As for the tissues related to carcass traits, such as subcutaneous adipose, abdominal adipose, longissimus dorsi muscle and leg muscle, the *SNX29* gene was also widely detected. Across these 11 tissues, the *SNX29* gene expression level in large intestine was markedly higher than other tissues, except for spleen (*p* < 0.05). The result suggested that *SNX29* could play an important role in the developmental process of XDB goats.

In addition, to explore for the relationship between *SNX29* CNV and the mRNA expression level of *SNX29* gene in goats. The mRNA expression of *SNX29* gene in abdominal adipose, subcutaneous adipose, longissimus dorsi muscle and leg muscle tissues of 30 female XDB goats was analyzed. The individuals with copy number loss type were not detected in these samples, which might be due to the low frequency of copy number loss type and limited samples. In these four tissues, the mRNA expression of goat *SNX29* gene was not significantly correlated with *SNX29* CNV types (*p* > 0.05). However, in longissimus dorsi muscle, individuals with *SNX29* copy number normal type had a higher trend for *SNX29* gene mRNA expression level than those with copy number gain type (*p* = 0.07), whereas individuals with *SNX29* copy number gain type had a higher trend in abdominal adipose (*p* = 0.09) (Figure 2B).

## 4. Discussion

Goat, as an important ruminant economic animal in human life, was distributed all over the world. With the development of human daily life’s requirements, the improvement of Chinese native goat production and meat quality performance has become the primary tasks of goat industry in China. Genetic variation occurs continuously in the biological world, mainly refers to genomic variation [26]. These variations might have effects on animal phenotype traits, which were particularly important for the evolution and continuation of animals. Currently, SV have attracted more attentions because of their potential applications in animal breeding due to their broader coverage [27,28].

In this study, to investigate the function of the *SNX29* gene, the protein structure and function of this gene were first analyzed. We found four conserved structural domains on SNX29 protein. Meanwhile, the sequence of *SNX29* CNV was overlapped with ZIP_TSC22D-like super family and DUF460 superfamily. The TSC22D structural domain contains proteins involved in a variety of functions such as cell proliferation, differentiation, development, and immune regulation [29,30]. As for the genetic variations of *SNX29* in this study, the 17 bp indel sequence was located at the front end of the *SNX29* gene, the 21 bp indel sequence was located in the intron region. Notably, *SNX29* CNV overlapped with a part of exons of *SNX29* gene. Thus, we hypothesized that *SNX29* CNV might affect the regulation of the transcriptional structure of the *SNX29* gene, thereby affecting the expression of this protein. Meanwhile, this study found that the *SNX29* protein motif and structure are highly conserved in domestic animals, including goats, which suggested that this gene could be stably inherited and might play important roles in animal growth and development.

In the previous study, the research on *SNX29* gene CNV was firstly carried out for African meat goats [18]. However, the studies on the genetic variation of *SNX29* gene have not been conducted in Chinese goat populations. Therefore, our study firstly validated the genetic variation of *SNX29* in Chinese local goats. The distribution proportion of copy number of normal type was the highest (62%), the proportion of loss type was the least (17%), and the proportion of gain type was 21% in this population. In the association study of CNV and mRNA expression of *SNX29*, the copy number of loss type was not detected in the randomly selected goats, which may be due to the small number of samples and low proportion of the loss type.

As for the CNV locus of the *SNX29* gene, it was found to be significantly correlated with XDB goat growth traits, especially CC and AC traits. This work found that the normal type of goats was superior in CC and AC traits. Therefore, in XDB goats, selecting the individuals with copy number normal type genotype frequency in the population can improve the growth performance of XDB goats. As for the two indels (17 bp indel and 21 bp indel), no significant effects of them on the growth traits of goat were observed in XDB goats (*p* > 0.05). However, in our previous study, the two indels were found to be significantly related to the growth traits such as chest width and chest depth in Shaanbei white cashmere goats [19]. Besides, compared with our previous studie [19], we found that 17 bp indel and 21 bp indel had significant differences in genotype and allele distribution between the two breeds.We speculated that these differences were due to the different genetic background among the two species. As known, the Shaanbei white cashmere goat is a dual-purpose goat breed known for wool and mutton production in north China, but the XDB goat is a local meat breed in south China. The body shape and development of goats were reflected by growth traits, which controlled by multiple genes [31]. Similarly, Zhang et al. [26] discovered the 14 bp indel and 17 bp indel of *SPAG17* gene were associated with the body size traits of Shaanbei white cashmere goat, but no significant association was identified in Hainan black goat. Hence, marker assisted selection (MAS) using the *SNX29* gene variations should be based on breed differences in goats.

The expression profile of *SNX29* showed high expression of *SNX29* in tissues such as the small intestine, spleen, and subcutaneous adipose. This finding revealed that the *SNX29* gene might play a crucial role in subcutaneous adipose deposition of XDB goat. It had been demonstrated that *SNX29* was associated with the developmental regulation of the nervous system in human [32]. To investigate the genetic effects of *SNX29* CNV at the level of gene transcription, we studied the correlation between *SNX29* CNV and mRNA expression. In this research, the relative expression of *SNX29* gene with copy number normal had higher level than that with copy number gain in muscle (*p* = 0.07), whereas the opposite trend was observed for abdominal adipose (*p* = 0.09). These results suggested that the gain type has a trend for inhibiting the *SNX29* gene expression in muscle and promoting the *SNX29* gene expression in adipose. Similarly, a study showed that *TSPY* copy number was negatively correlated with its mRNA expression in the testis of Canadian Holstein Bulls [33]. Wang et al. [34] reported that a weak negative correlation was shown between *RHACD8* mRNA levels and *RHACD8* copy number in the chicken. Gene expression could be potentially affected by CNV through changing in gene dose, and transcriptional structure [35]. In addition, CNV could change gene expression through position effect, fusion effect, gene blocking and other pathways, thus affecting phenotypic traits [36]. Previous studies have shown that *SNX29* is an important candidate gene related to the slaughter traits (e.g., backfat thickness) and meat quality traits (e.g., content of intramuscular fat) in pigs [17]. Taking these findings together, we speculated that the *SNX29* CNV might influence the growth traits of XDB goats by affecting the *SNX29* gene expression levels in muscles and adipose. However, the molecular regulatory mechanism of *SNX29* genetic variations affecting goat growth traits is needed to clarify in further study. Also, the results of this study need to be further studied in more breeds and samples.

## 5. Conclusions

In summary, three SV of *SNX29* gene (two indels and a large CNV) were firstly investigated in local meat goats of Hunan Province, including the distributions, the genetic effects on *SNX29* gene expression and growth traits in XDB goats. The results showed that the CNV was significantly associated with growth traits. Our study indicated the genetic variations of *SNX29* gene might play a role in the growth and development of XDB goat, and the *SNX29* gene CNV could be used as a molecular marker for the early MAS of XDB goat breeding.

## Figures and Tables

**Figure 1 animals-12-03461-f001:**
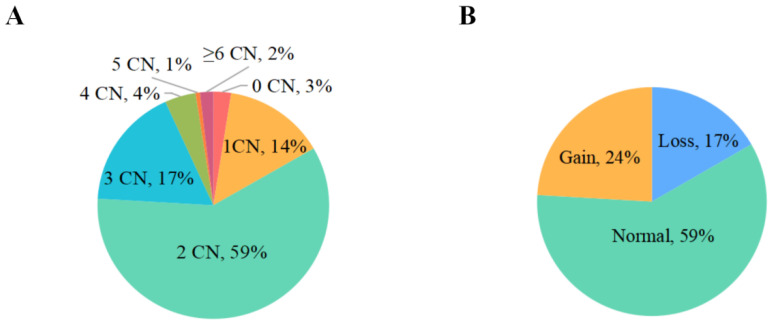
Distribution of copy number variation of the *SNX29* in Xiangdong black goats. (**A**) The frequency of the copy numbers of the SNX29 CNV in Xiangdong black goats. The number of 0 to ≥6 represented the copy number (CN) of SNX29 CNV; (**B**) Distribution of different CNV types in Xiangdong black goats. Loss, Normal, Gain was defined as copy number <2, =2, or >2, respectively.

**Figure 2 animals-12-03461-f002:**
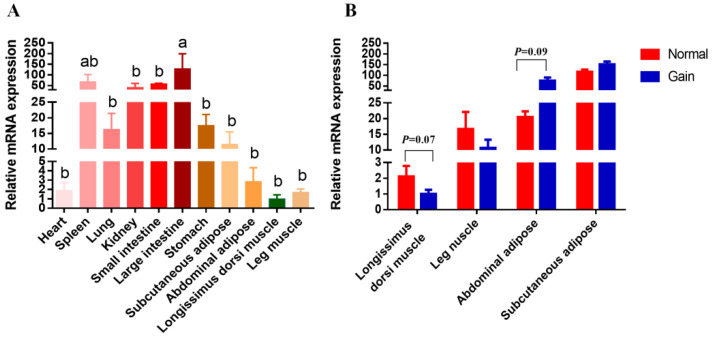
Comparison analysis of SNX29 gene expression levels among different tissues (**A**) and different CNV genotypes (**B**) in Xiangdong black goats. (**A**) The *SNX29* gene mRNA expression profile in 11 different tissues of three adult female XDB goats. The relative mRNA expression level of *SNX2*9 gene were calculated by 2^−△△Ct^ method. The *GAPDH* was used as the internal reference gene. Different letters (a, b) within the same row represent significant differences among the three groups. (**B**) Comparison of *SNX29* gene expression levels among different *SNX29* CNV genotypes in abdominal adipose, subcutaneous adipose, longissimus dorsi muscle, and leg muscle tissues of 30 female XDB goats.

**Table 1 animals-12-03461-t001:** Relationship of copy number of *SNX29* CNV with growth traits in Xiangdong black goats.

	CNV Types (Mean ± SE)			
Growth Traits	Loss (*n* = 88)	Normal (*n* = 304)	Gain (*n* = 124)	*p* Value
Age (day)	738.48 ± 17.48	732.96 ± 16.73	721.36 ± 16.29	0.281
Body height (BH)	55.30 ± 0.98	56.30 ± 0.44	56.76 ± 0.65	0.407
Body length (BL)	57.96 ± 0.96	59.47 ± 0.51	58.09 ± 1.11	0.262
Body oblique length (OL)	58.48 ± 1.41	59.89 ± 0.47	58.52 ± 1.11	0.278
Chest circumference (CC)	72.04 ± 1.50 ^a,b^	74.89 ± 0.72 ^a^	70.09 ± 1.42 ^b^	0.004 **
Abdominal circumference (AC)	81.56 ± 1.91 ^b^	86.50 ± 1.01 ^a^	81.24 ± 1.62 ^b^	0.007 **
Cross height (CH)	55.96 ± 0.96	57.80 ± 0.39	56.91 ± 0.70	0.103
Cannon Circumference (CAC)	11.05 ± 0.30	10.80 ± 0.11	10.61 ± 0.21	0.216

Note: Data were shown as mean ± SE. Values with different letters (^a,b^) within the same row represent significant differences among the three groups. ** indicates the values differ significantly at *p* < 0.01.

## Data Availability

Not applicable.

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
