# Peer review of "Effects of Genetic Variation of the Sorting Nexin 29 (SNX29) Gene on Growth Traits of Xiangdong Black Goat"

_animals, 2022, doi:10.3390/ani12243461_

Round 1

Reviewer 1 Report (New Reviewer)

The manuscript presented by Chen et al. characterized three structural variations (SV) of the sorting nexin 29 (SNX29) gene, including two small insertion and deletions (indels) and a large copy number variation (CNV), in Xiangdong black goats. The study demonstrated that the CNV was significantly associated with chest circumference and abdominal circumference of the studied goat population. I have some major and minor comments, see below:

Major comments:

Genomic variation could be, according to their size, classified as single nucleotide polymorphism (SNP, i.e. single base change), insertion and deletion (indel, i.e. small structural variation with size ranging from 2-50nt), structural variation (SV, i.e. large structural variation with size > 50nt). Essentially, the later two types are genomic structural variation, and copy number variation is just a type of structural variation, reflecting duplication (copy number gain) and deletion (copy number loss). It should be noted that structural variation also contains chromosomal inversion. In light of above mentioned, I advise authors to revise concepts throughout the manuscript. For instance, line 52-53.

Minor comments:

1.            Following the above, when specifying two indels with their size, using “17bp-indel and 21bp-indel” e.g. line 33, is uninformative. Including their states, e.g. “17bp-deletion and 21bp-deletion” when compared to reference sequence, will be more informative.

2.            Figure 1 resolution is two low

3.            Please provide more details about how the tree in Figure 1C is constructed in figure legend, how is this tree when compared to species tree?

4.            In line 179, how you identify motifs? Using MEME in line 97? I can not locate the corresponding method? What are predicted functions of the RUN, ZIP_TSC22D-like super family, DUF460 super family and PX_RUN?

5.            Figures 1D and 1E are unclear. Is the first arrow with blue-white color gene? Is the second light green-white arrow mRNA? Why the second is larger than the first? Where are two indel locations? You can specify name and location on the left side, and plot bar on right side. If the entire Figure 1E is the second blue bin in Figure 1D, or just part of it (two black line indicated) is the second blue bin in Figure 1D?

6.            Line 118, what is the name of “the kit (Takara, Japan)”?

7.            Line 122, why you use MC1R as internal reference? I think this is because caprine MC1R was supposed to be a single-copy gene (i.e. no copy number variable), you should cite relevant paper?

8.            Line 251, change “means” to “mean” when it means “average”, please do same throughout the manuscript (including supplemental tables).

9.            When you mentioned 2−ΔΔCt method, e.g. lines 155, 265, please cite “Analysis of relative gene expression data using real-time quantitative PCR and the 2−ΔΔCT method (10.1006/meth.2001.1262)”

10.          In Discussion section, you could include some description about the frequency difference of the two indels when compared to your previous publication: [19].

11.          Reference format does not meet the requirement of the journal (https://www.mdpi.com/journal/animals/instructions#references), e.g. YEAR should be after journal name

12.          language editing with native speakers is required.

Author Response

Reviewer 1:

The manuscript presented by Chen et al. characterized three structural variations (SV) of the sorting nexin 29 (SNX29) gene, including two small insertion and deletions (indels) and a large copy number variation (CNV), in Xiangdong black goats. The study demonstrated that the CNV was significantly associated with chest circumference and abdominal circumference of the studied goat population. I have some major and minor comments, see below:

Major comments:

Genomic variation could be, according to their size, classified as single nucleotide polymorphism (SNP, i.e. single base change), insertion and deletion (indel, i.e. small structural variation with size ranging from 2-50nt), structural variation (SV, i.e. large structural variation with size > 50nt). Essentially, the later two types are genomic structural variation, and copy number variation is just a type of structural variation, reflecting duplication (copy number gain) and deletion (copy number loss). It should be noted that structural variation also contains chromosomal inversion. In light of above mentioned, I advise authors to revise concepts throughout the manuscript. For instance, lines 52-53.

Respond: Thank you very much for your kind and professional comments. We agree with your suggestions. In the revised manuscript, we have revise the concepts throughout the manuscript. Please see lines 53-59.  

Comment 1-1:

Following the above, when specifying two indels with their size, using “17bp-indel and 21bp-indel” e.g. line 33, is uninformative. Including their states, e.g. “17bp-deletion and 21bp-deletion” when compared to reference sequence, will be more informative.

Respond 1-1:

Thank you for this valuable point. We have added two indels related information in lines 33-34.

Comment 1-2:

Figure 1 resolution is two low

Respond 1-2:

We have improved the clarity of this Figure. As suggested by Reviewer 2, we have moved the previous Figure 1 to Supplementary Figures part. Please find it in the Figure S1 in the revisions.

Comment 1-3:

Please provide more details about how the tree in Figure 1C is constructed in figure legend, how is this tree when compared to species tree?

Respond 1-3:

Thank you for this suggestion. We have added the details on the constructed method for this Figure in the Figure S1 legend. The amino acid sequences of the SNX29 protein for 11 species were downloaded from the NCBI database(https://www.ncbi.nlm.nih.gov/protein). Phylogenetic tree analysis for SNX29 protein among different species were performed by MUSCLE neighbor-joining (NJ) method with MEGA X 10.2.6 software. In this study, the phylogenetic tree was constructed based on the SNX29 gene protein sequences across different species and could indicate the evolutionary history in terms of the SNX29 gene. As for the species tree, the whole genome sequences shall be used for analysis. 

Comment 1-4:

In line 179, how you identify motifs? Using MEME in line 97? I can not locate the corresponding method? What are predicted functions of the RUN, ZIP_TSC22D-like super family, DUF460 super family and PX_RUN?

Respond 1-4:

Thank you for pointing this out. We apologize for the unclear description. The motifs of SNX29 proteins among different species were investigated by the MEME suite. We have revised the descriptions in lines 106-108. The predictive function of conservative domains were analyzed by CDD from NCBI (https://www.ncbi.nlm.nih.gov/cdd/) (line 110, lines 199-200).

Comment 1-5:

 Figures 1D and 1E are unclear. Is the first arrow with blue-white color gene? Is the second light green-white arrow mRNA? Why the second is larger than the first? Where are two indel locations? You can specify name and location on the left side, and plot bar on right side. If the entire Figure 1E is the second blue bin in Figure 1D, or just part of it (two black line indicated) is the second blue bin in Figure 1D?

Respond 1-5:

Thank you for pointing this out. We apologize for the error in the previous version. We have revised this Figure to make it easier to be understood. Meanwhile, we have added the marks for the positions of two indels in the Figure S1D. Figure S1D showed the location of SNX29 CNV and two indels in goat SNX29 gene DNA region and mRNA region, especially based on NC_030832.1 (Chr25: 10572059-10683032) and XM_018040633.1/XM_018040634.1. For the two indels, the 17bp indel (g.10559298-10559314) sequence was located in the upstream of the SNX29 gene and the 21bp indel (g.10918982-10919002) sequence was located in the intron region. SNX29 CNV was overlapped with several extrons and introns of SNX29 gene both in the two predicted transcripts (XM_018040633.1 and XM_018040634.1). Also, as shown in Figure S1E, the SNX29 CNV - related protein region was related with a part of conserved domains in SNX29 protein sequence (XP_017896122.1/XP_017896123.1) based on NCBI CDD. Please see Figure S1D and 1E for details.

Comment 1-6:

Line 118, what is the name of“the kit (Takara, Japan)”?

Respond 1-6:

Thank you for pointing out this inadequate description. We have modified the description in lines 133-134: “Then, RNA was synthesized to first-strand cDNA using a PrimeScriptTM RT reagent Kit with gDNA Eraser (Takara, Japan) according to the instruction manufactory.”

Comment 1-7: 

Line 122, why you use MC1R as internal reference? I think this is because caprine MC1R was supposed to be a single-copy gene (i.e. no copy number variable), you should cite relevant paper?

Respond 1-7:

Thanks for your comments. MC1R has been widely used as the reference gene in previous studies such as (Li et al., 2020) and (Liu et al., 2021). We have added the reference (Liu et al., 2021) in the revised version (line 139, lines 431-432). 

Reference:

Li L, Yang P, Shi S, Zhang Z, Shi Q, Xu J, He H, Lei C, Wang E, Chen H, Huang Y. Association Analysis to Copy Number Variation (CNV) of Opn4 Gene with Growth Traits of Goats. Animals (Basel). 2020;10(3):441.

Liu M, Cheng J, Chen Y, Yang L, Raza SHA, Huang Y, Lei C, Liu GE, Lan X, Chen H. Distribution of DGAT1 copy number variation in Chinese goats and its associations with milk production traits. Animal Biotechnol. 2021;1-6.

Comment 1-8:

 Line 251, change “means” to “mean” when it means “average”, please do same throughout the manuscript (including supplemental tables).

Respond 1-8:

Thank you very much for pointing out this mistake. As suggested, we have changed "means" to "mean" in this study (line 247).

Comment 1-9: 

 When you mentioned 2−ΔΔCt method, e.g. lines 155, 265, please cite “Analysis of relative gene expression data using real-time quantitative PCR and the  2−ΔΔCt method (10.1006/meth.2001.1262)”

Respond 1-9:

As suggested, we have added references in the relevant sections (line 173, lines 433-434).

Comment 1-10: 

In Discussion section, you could include some description about the frequency difference of the two indels when compared to your previous publication: [19].

Respond 1-10:

Thanks for your good suggestion. We have added a discussion section based on your comments (lines 322-324).                                                                                                                                                                                                                                                                                                                                                                                                                                                                                                                                                                                                                              

Comment 1-11: 

Reference format does not meet the requirement of the journal (https://www.mdpi.com/journal/animals/instructions#references), e.g. YEAR should be after journal name

Respond 1-11:

Thank you very much for pointing out these errors. In this revised manuscript, we checked all the references and corrected these references’ style.

Comment 1-12:

 language editing with native speakers is required.

Respond 1-12:

Thank you for your kind suggestion. We have carefully checked and revised the manuscript. Also, we asked native English speakers for help, to improve the English writing in our manuscript. 

Reviewer 2 Report (New Reviewer)

The manuscript is well presented and only minor English editing problems need to be improved.

Author Response

Reviewer 2:

Comment 1-1:

The manuscript is well presented and only minor English editing problems need to be improved.

Respond 1-1:

Many thanks for the reviewer #2's constructive comments and suggestions. We have carefully checked and improved the English writing in the revised manuscript. Also, we asked native English speakers for help, to improve the English writing in our manuscript.

Reviewer 3 Report (New Reviewer)

Dear Authors,

I read your article with interest. Although the topic has some interest, I believe that some parts of the report need to be revised. For this reason I attach the paper with the notes directly reported in the text.

Author Response

Reviewer 3:

Dear Authors,

I read your article with interest. Although the topic has some interest, I believe that some parts of the report need to be revised. For this reason I attach the paper with the notes directly reported in the text.

Respond: Thank you very much for your carefully, valuable and thoughtful comments. We have responded to your questions and suggestions in the document one by one throughout the manuscript. Please find our answers to your comments one by one in the attached files. We hope our revisions will make sense.

Round 2

Reviewer 3 Report (New Reviewer)

Dear Authors,

I have read your second version and I am very satisfied with the corrections you have made.

I only ask you to consider the small notes that I have inserted directly in the paper, especially as regards the age of the animals included in the three groups of Table 1: this is very important information when reporting growth values.

Best Regards

Author Response

Dear Authors,

I have read your second version and I am very satisfied with the corrections you have made.

I only ask you to consider the small notes that I have inserted directly in the paper, especially as regards the age of the animals included in the three groups of Table 1: this is very important information when reporting growth values.

Respond: Many thanks for the reviewer #3's constructive comments and suggestions, which enabled us to greatly improve the quality of our manuscript. In this revised manuscript, we have revised the paper based on those insightful suggestions. Please find our answers to your comments one by one in the attached files. We hope our revisions will make sense.

This manuscript is a resubmission of an earlier submission. The following is a list of the peer review reports and author responses from that submission.

Round 1

Reviewer 1 Report

In this study, two indels (17bp-indel and 21bp-indel) and CNV were identified in 516 individual Xiangdong black goats (a representative Chinese native goat breed), while SNX29 CNV was significantly associated with chest circumference and abdominal circumference. And SNX29 was widely expressed in tissues, especially in small intestine, spleen, and subcutaneous fat. These findings suggested that the SNX29 gene could be used for marker-assisted selection in meat goats. There are some issues that need to be addressed before publication.

1.      Introduction: The gene expression and regulatory pathway of SNX29 should be descripted to explain the potential association with growth traits.

2.      Ln 66: no references listed above suggested the SNX29 could be related to reproduction traits. Please supplement the corresponding references.

3.      Ln 97: The 516 adult female goats were all nonpregnant or pregnant?

4.      Ln 149 and 152: Please supplement the references for the calculation method.

5.      Ln166: The sentence should be revised.

6.      Ln 173 and 177: The description about methods should be canceled, or move to the Methods section.

7.      Figure 5 and Figure 6: The sample number of each group should be given in the legend.

8.      All data were given as mean±SD on line 153, but the data were mean±SE in Table 1, and were LSM±SE in Table 3, which should be confirmed.

9.      Table 1 and 3: the note, A, B/ a, b described in the annotation stand for P< 0.01 / P<  0.05, but there are only A, B and * marks in the table, please carefully check the description.

10.   Figure 5 and Figure 6: The calculation methods were same, why the values of relative mRNA expression (Y-axis) were very different.

11.   Ln 327-328: There no results supported the SNX29 might have roue in reproduction of livestock.

12.   Ln 339: No sufficient evidence supported the hypothesis that the loss type of SNX29 might inhibit muscle development.

13. Ln 346: The “first time” should be reconsider. 

Reviewer 2 Report

    1.   Overemphasizing the gene and the results.

    2.   Need references and/or links for software packages.

    3.   Mix of tables 1↔2.

    4.   Inconsistent format of tables.

    5.   Detailed legends are needed.

    6.   Need to start a result section with a short reminder of what was done (exactly what analysis created this result, Methods section number included).

    7.   Needed are few reminders that only females were used.

    8.   Present numerical data of the figures.

    9.   Present the species where the quoted study took place.

10.   Present the statistical procedure used for each figure and table.